# MasHost Builds It All: Autonomous Multi-Agent System Directed by Reinforcement Learning

## Abstract

Large Language Model (LLM)-driven Multi-agent systems (Mas) have recently emerged as a powerful paradigm for tackling complex real-world tasks. However, existing Mas design strategies typically rely on manually crafted interaction mechanisms or heuristic rules, introducing human biases and constraining the autonomous ability. Even recent advances that claim to adaptively construct Mas still fall within the paradigm of semi-autonomous patterns. In this work, we introduce `MasHost`, a reinforcement learning (RL)-based framework designed for autonomous and query-adaptive Mas generation. Firstly, we formulate the generation of Mas as a graph search problem and propose Hierarchical Relative Policy Optimization (HRPO), a novel RL strategy that collaboratively combines group-level relative advantages with fine-grained action-wise rewards. Secondly, we design `MasHost` to jointly sample agent roles and their interactions through a unified probabilistic sampling mechanism, enabling adaptive and coherent Mas construction. Beyond the conventional emphasis on accuracy and efficiency, `MasHost` innovatively introduces component rationality, offering a new perspective on the principled design of multi-agent systems. To our knowledge, our proposed `MasHost` is the first RL-driven framework for autonomous Mas graph construction. Extensive experiments on six benchmarks demonstrate that `MasHost` consistently outperforms most competitive baselines, validating its effectiveness, efficiency, and structure rationality.

## 1 Introduction

**Background.** In recent years, the advent of large language models (LLMs) has fundamentally reshaped research paradigms across various fields Achiam et al. (2023); Shao et al. (2024); OpenAI (2024). LLM-driven Multi-agent system (Mas) demonstrate remarkable potential in addressing complex real-world tasks, emerging as a prominent research frontier in artificial intelligence Zhuge et al. (2024); Zhang et al. (2024a;b); Wang et al. (2023); Yang et al. (2023); Hong et al. (2023); Liu et al. (2023); Wu et al. (2023); Ye et al. (2025); Zhang et al. (2025); Hu et al. (2024); Chen et al. (2023). Mas seeks to address tasks that surpass the capabilities of a single agent through coordinated interactions among multiple agents Luo et al. (2025); Li et al. (2024c); Guo et al. (2024). Therefore, designing the interaction mechanism among agents is critical to ensuring the effectiveness of Mas. Many studies rely on manual drafting and heuristic-based approaches for constructing interaction mechanisms Wei et al. (2022); Wang et al. (2023); Du et al. (2023); Wang et al. (2022). However, these strategies often yield suboptimal performance due to the introduction of human biases.

**Motivation.** This limitation has prompted recent efforts toward the development of autonomous Mas. These works model Mas as a directed graph to achieve policy-driven Mas construction, facilitating more adaptive and flexible connections among agents Zhuge et al. (2024); Zhang et al. (2024a;b); Hong et al. (2023); Hu et al. (2024); Chen et al. (2023); Yue et al. (2025); Zhang et al. (2025). Despite these advances, full autonomous Mas remains elusive. ❶ **Candidate Pool Sampling** strategy is followed by many existing approaches Zhang et al. (2025); Yue et al. (2025); Chen et al. (2023), where Mas are constructed by sampling or composing from a predefined structure pool. This candidate pool inevitably introduces human biases, limiting the flexibility of model in Mas design. ❷ **Agentic Workflow** is also a widely adopted strategy in prior works Zhang et al. (2024b;a);

Zhuge et al. (2024); Hu et al. (2024), aiming at the design of task-level workflows through an adaptive method. These workflows exhibit limited adaptability across varying in-task queries, which often results in suboptimal trade-offs between performance and cost-efficiency. Therefore, existing methods remain within the realm of semi-autonomous design.

**Insight.** We argue that the constrained search spaces in recent practices fundamentally restrict the autonomous ability of Mas. Candidate pool sampling limits the search space of Mas due to predefined structure pool, whereas agentic workflows inherently constrain the Mas search to a coarse granularity at the task level. To overcome these limitations, we aim to model the Mas construct process over the full-scale graph search space, enabling fully autonomous and query-adaptive Mas design. However, implementing a full-scale graph search to construct autonomous Mas presents significant challenges. The primary challenge stems from the non-Euclidean nature of the Mas graph, where the expansive combinatorial space of node feature sampling and edge learning complicates the modeling and optimization process.

**Methodology.** In this work, we propose an autonomous Mas hosting framework (`MasHost`) based on Reinforcement Learning (RL) algorithm. This design is motivated by the recognition that RL strategy can effectively optimize the exploration of vast search spaces, supported by numerous successful applications Sun et al. (2024); Kaelbling et al. (1996); Li (2017). Specifically, we model the design of Mas as a graph construction process from scratch under RL guidance. **Firstly**, the challenge lies in the dual-decision nature of the Mas construction process, which involves both node role generation and connectivity decision. This differs fundamentally from conventional RL algorithms designed for single-step or sequential actions. Discretizing

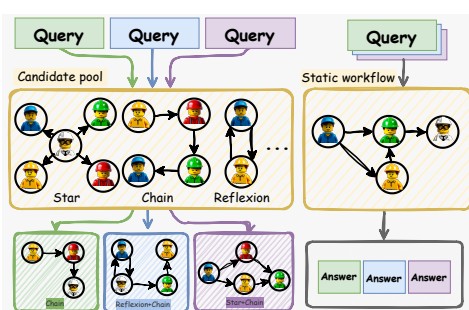

Figure 1: *(left)* Candidate Pool Sampling Mas. *(right)* Agentic Workflow.

this dual-action process not only introduces convergence difficulties in high-dimensional combinatorial spaces but also disrupts gradient flow. To address this, we propose a joint probabilistic sampling mechanism that simultaneously models the distribution over agent attributes and their connectivity patterns. Technically, we sample agent roles from the full-scale role space, and subsequently guide the connectivity decisions using joint residual probabilities derived from the role assignments. This mechanism not only ensures efficient representation of the Mas design process but also enables the optimization of differentiable sampling. **Secondly**, the next challenge remains in formulating an effective RL objective that aligns with the autonomous Mas construction paradigm. This difficulty arises from the fact that our Mas construction is driven by three objectives. Beyond the performance and efficiency goals emphasized in prior Mas works, we place additional attention on ensuring the structure rationality of the constructed systems. To achieve this, we propose a novel RL optimization pipeline, Hierarchical Relative Policy Optimization (HRPO), which enables policy-driven Mas to respond to queries accurately, efficiently, and rationally. Inspired by GRPO Shao et al. (2024), HRPO incorporates a hierarchical reward structure that combines group-relative advantages with action-wise absolute rewards. The group-relative advantage strategy compares the relative performance of different Mas, guiding the policy network to prioritize accuracy and efficiency in query responses from well-performing Mas. The step-wise absolute reward emphasizes the rationality of each action, ensuring that the addition or removal of each agent aligns with the overall objective. **Finally**, we conduct comprehensive experiments focusing on three aspects, i.e., performance, cost-efficiency, and rationality. Empirical comparisons of accuracy and cost-effectiveness with existing methods verify the effectiveness of `MasHost`. Our **contributions** can be summarized as:

- We introduce a reinforcement learning-enhanced framework for multi-agent system design, enabling fully autonomous agent generation from scratch.

- We propose a joint probabilistic sampling mechanism to realize the dual-action process in Mas construction, along with a hierarchical relative policy optimization algorithm to optimize the system for high performance, efficiency, and rationality.

- Extensive experiments on six benchmarks demonstrate that `MasHost` consistently outperforms most competitive baselines, validating its effectiveness, efficiency, and structural rationality.

## 2 PRELIMINARY

### 2.1 GRAPH FOR MULTI-AGENT SYSTEM

Modeling Multi-agent systems (Mas) as directed graphs $\mathcal{G} = (\mathcal{V}, \mathcal{E})$ has become a prevailing paradigm in recent researches. Each node $v \in \mathcal{V}$ represents an LLM agent with role-specific attributes that include its capabilities and responsibilities, while each directed edge $e \in \mathcal{E}$ encodes an interaction pathway between agents. This formulation offers a flexible and generalizable abstraction for Mas, and recent efforts have advanced this paradigm to design autonomous Mas architectures for tackling real-world applications.

### 2.2 REINFORCEMENT LEARNING FOR MULTI-AGENT SYSTEM

We formulate the Mas construction process as a Markov Decision Process $\mathcal{M} = (\mathcal{S}, \mathcal{A}, \mathcal{R})$.

- **State Space $\mathcal{S}$.** The state space $\mathcal{S}$ covers the global configuration of the Mas. At step $t$, the state $s_t \in \mathcal{S}$ encapsulates the query $Q$, constructed structure $\mathtt{M}_t = \{\mathtt{R}_1, ..., \mathtt{R}_{|\mathtt{M}_t|}\}$, and the message list of those agents $\mathrm{MESSAGE}(\mathtt{M}_t)$, i.e., $s_t = \{Q, \mathtt{M}_t, \mathrm{MESSAGE}(\mathtt{M}_t)\}$. Moreover, the output of each agent $\mathtt{R}_j$ can be formalized as $\mathrm{MESSAGE}(\mathtt{R}_j)$, where $j \in [1, |\mathtt{M}_t|]$.

- **Action Space $\mathcal{A}$.** The action space $\mathcal{A}$ defines all possible editing operations for constructing the Mas from scratch. It consists of two categories: node-level actions $\mathcal{A}_n$ and edge-level actions $\mathcal{A}_e$.

- **Policy Function $\pi$.** The policy function $\pi$ governs the decision-making process of Mas construction by jointly modeling node-level and edge-level actions.

- **Reward Function $r(\mathtt{a}_t)$.** The reward function $r(\mathtt{a}_t)$ defines the reward of each action $\mathtt{a}_t \in \mathcal{A}$ taken in a given state $s \in \mathcal{S}$. To achieve stabilize policy optimization, the advantage function $A(\mathtt{a}_t)$ is commonly introduced, which quantifies the relative merit of an action by measuring the difference between the action's expected return and the baseline value of the current state.

Building on the above understanding, the construction of Mas can be formulated within the RL paradigm as a sequence of state-action transitions, represented as $(s_0, \mathtt{a}_1, s_1, \mathtt{a}_2, s_2, \cdots)$, where each state $s_t$ corresponds to the current configuration of the Mas, and each action $\mathtt{a}_t$ represents an editing operation that transitions the system from one state to the next. A comprehensive account of the design motivations and strategies for the above concepts will be provided in Sec. 4.

### 2.3 PROBLEM FORMULATION

Given a query $Q$, we focus on leveraging RL policy to learn an optimal $\pi^* = (\pi_\theta^*, \pi_\phi^*)$ for Mas design, enabling fully autonomous and query-specified construction of multi-agent systems. We define the optimality of Mas $\mathtt{M}$ from three perspectives: performance quality, resource efficiency, and the structure rationality. Therefore, the overall reward function $\mathcal{R}(\mathtt{M} \mid Q)$ is formulated as a composition of three key criteria,

$$r(\mathtt{M} \mid Q) = r_{\mathrm{perf}}(\mathtt{M}, Q) + r_{\mathrm{eff}}(\mathtt{M}, Q) + r_{\mathrm{struct}}(\mathtt{M}). \tag{1}$$

where $r_{\mathrm{perf}}(\mathtt{M}, Q)$ measures performance quality in answering query, $r_{\mathrm{eff}}(\mathtt{M}, Q)$ evaluates resource efficiency in answering query, and $r_{\mathrm{struct}}(\mathtt{M})$ captures structure rationality. The objective is to find $\pi^*$ that maximizes the expected reward,

$$\pi^* = \arg\max_\pi \mathbb{E}_{\mathtt{M} \sim \pi} \left[ r(\mathtt{M} \mid Q) \right]. \tag{2}$$

## 3 RELATED WORK

In recent years, the emergence of Large Language Models (LLMs) has introduced new research paradigms for tasks such as mathematical reasoning, code generation, data analysis, and question answering Shao et al. (2024); Li et al. (2024a); Zhu et al. (2024); Xie et al. (2024); Song et al. (2023); Wang et al. (2024); Zha et al. (2023). Empirical studies have further shown that challenges unsolved by a single LLM can be effectively addressed through collaborative interactions among multiple LLM-based agents with specialized roles Wei et al. (2022); Yao et al. (2023); Shinn et al.

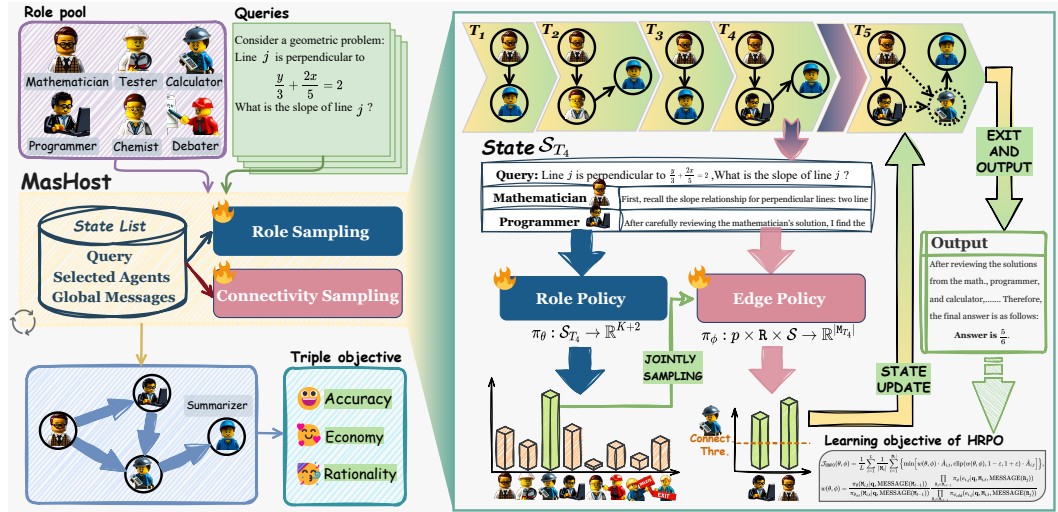

Figure 2: Framework of our `MasHost`. *(left)* `MasHost` autonomously manages the complete process of building Mas. *(right)* Detailed construction of Mas using a reinforcement learning strategy.

(2023), giving rise to the development of Multi-agent systems (Mas). Various Mas patterns have been explored, including chain-based, star-shaped, debate-style, and tree-structured frameworksWei et al. (2022); Zhou et al. (2024); Du et al. (2023); Ishibashi & Nishimura (2024); Li et al. (2024b), leading to notable successes across diverse domains.

**Agentic Workflow.** Workflow-based approaches statically perform tasks by following predefined workflows, which is implemented by multiple agents. Designing workflows based on handcraft design and learnable network constitute two prominent application paradigms. The former aims to design workflows based on human understanding and domain knowledge, such as code generation Ridnik et al. (2024), mathematics Deng & Mineiro (2024); Zhong et al. (2024), and question answering Nori et al. (2023). The latter focuses on the automated construction of workflows, where an adaptive algorithm can dynamically design all task-specific workflows. GPTSwarm Zhuge et al. (2024) models workflows as graphs, and leverages reinforcement learning to design task-specific workflows. ADAS Hu et al. (2024) represents workflows using code structures and maintains historical workflows in a linear list. AFLOW Zhang et al. (2024b) also represents workflows through code, emphasizing a custom MCTS algorithm for automated workflow optimization.

**Autonomous Mas.** Different from workflow-based practices, autonomous Mas efforts focus on designing the most efficient and accurate Mas tailored to each query. MaAS Zhang et al. (2025) constructs Mas by building an agentic supernet, where each block within the supernet is sampled from a predefined structure pool. MasRouter Yue et al. (2025) constructs Mas by sampling from four structure candidate pools while adaptively learning the number of agents, role types, and LLM types. MAS-GPT Ye et al. (2025) represents Mas as executable code and trains a LLM to construct Mas by generating code. Actually, existing approaches remain semi-autonomous. The reason lies that most methods model Mas construction as sampling or combining from predefined structure pools. Even for the seemingly fully autonomous framework MAS-GPT, the datasets used to train the LLM are still manually curated rather than generated through exploratory processes. Our work differs fundamentally from existing approaches by employing reinforcement learning to autonomously explore optimal Mas structures from scratch. This design enables the constructed Mas to be free from human biases and solely optimized for better query answering.

## 4 MASHOST: THE HOST FOR MULTI-AGENT SYSTEMS

The design of Mas involves a complex search space that encompasses both node attributes (e.g., agent roles) and connectivity patterns (e.g., inter-agent coordination). As a result, each step of the RL search process exhibits dual-action characteristics. To facilitate efficient search and ensure gradient differentiability, we introduce a Joint Probabilistic Space Sampling (JPSS) mechanism in Sec. 4.1. We then analyze the construction objectives in existing Mas studies and extend them in our

framework from three dimensions. To achieve this goal, we propose a novel Hierarchical Relative Policy Optimization pipeline specifically designed for agent system construction in Sec. 4.2. The implementation details of the network are described in detail in Fig. 5.

## 4.1 JOINT PROBABILITY SPACE SAMPLING

The action space $\mathcal{A}$ encompasses all editing operations for constructing Mas from scratch, comprising node-level actions $\mathcal{A}_n$ and edge-level actions $\mathcal{A}_e$. Therefore, the atomic action at time step $t$ is represented as a tuple of two sub-actions, $\mathtt{a}_t = (a_n, a_e)$. Our policy network $\pi$ consists of two components: the first policy $\pi_\theta$ selects actions from the space $\mathcal{A}_n$, and the secondary policy $\pi_\phi$ conducts the link decision from the space $\mathcal{A}_e$. At the step $t$, the action space $\mathcal{A}_n$ is modeled with three types of atomic actions $a_n \in \{\mathtt{ADD}, \mathtt{DELTE}, \mathtt{EXIT}\}$.

- $\mathtt{ADD}$. This action involves adding a new agent. Once triggered, a agent role is subsequently sampled from the role space $\mathcal{R} = \{R_1, ..., R_K\}$. Thus, the $\mathtt{ADD}$ action serves both as an activation signal and as a role sapling. In the implementation, we omit its function as an agent-adding signal and instead integrate role selection directly into the policy $\pi_\theta$. In other words, the single $\mathtt{ADD}$ action is replaced by the role space $\mathtt{ADD} := \mathcal{R}$.
- $\mathtt{DELTE}$. This action corresponds to removing an agent that was most recently modified.
- $\mathtt{EXIT}$. This action marks the completion of Mas construction, where the intermediate inference results are passed to a final summary agent, which then produces the answer to the query.

Given the above analysis of actions, the sampling space of $\pi_\theta$ can be defined as the union of the role space $\mathcal{R}$ and the special actions $\{\mathtt{DELETE}, \mathtt{EXIT}\}$, i.e., $\mathcal{A}_n = \mathcal{R} \cup \{\mathtt{DELETE}, \mathtt{EXIT}\}$, where $|\mathcal{A}_n| = K + 2$. Given the already constructed Mas $\mathtt{M}_t$ prior to step $t$, the policy network $\pi_\theta$ conducts sampling process with the sate $s_t$ as input, i.e., $a_n \sim \pi_\theta(\mathcal{A}_n|s_t)$.

Once the sampled action satisfies $a_n = \mathtt{R}_t \in \mathcal{R}$, the policy $\pi_\phi$ is activated. The policy network $\pi_\phi$ is designed to learn the interaction patterns $a_e$ between the newly added agent $\mathtt{R}_t$ and the existing agents $\mathtt{M}_t = \{\mathtt{R}_1, ..., \mathtt{R}_{|\mathtt{M}_t|}\}$. Technically, $\pi_\phi$ performs connectivity sampling using the current state $s_t$ and the selected role $\mathtt{R}_t$ as inputs, $a_e \sim \pi_\phi(\mathcal{A}_e|s_t, \mathtt{R}_t)$.

The independent learning of $\pi_\theta$ and $\pi_\phi$ is infeasible, which brings the issue of gradient disruption. To this end, we introduce the JPSS to effectively guide the dual-action decision process in Mas design. In the setting of JPSS, the process of constructing a Mas $\mathtt{M}$ based on RL is modeled by a unified policy procedure. Technically, $\pi_\theta$ is parameterized to produce a softmax distribution $\mathbb{P}_{a_n} \in \mathbb{R}^{K+2}$ over the role space $\mathcal{A}_n$, and sample the role $\mathtt{R}$ with the highest probability. Subsequently, $\pi_\phi$ takes $\mathtt{R}$ as input and outputs a sigmoid-based edge sampling distribution $\mathbb{P}_{a_e} \in \mathbb{R}^{|\mathtt{M}_t|}$. Instead of sampling directly from the probability distribution of $\mathbb{P}_{a_e}$, we conduct connectivity sampling based on the joint probability $a_e \sim p \times \mathbb{P}_{a_e}$, where $p$ denotes the probability of selecting $\mathtt{R}$. Under this setup, role selecting and connection learning are modeled as a unified action sampling $\mathtt{a}_t = (a_n, a_e) \sim \pi_\theta \times \pi_\phi$,

$$\pi_\theta : \mathcal{S} \to \mathbb{R}^K, \quad \pi_\phi : p \times \mathtt{R} \times \mathcal{S} \to \mathbb{R}^{|\mathtt{M}_t|}. \tag{3}$$

## 4.2 HIERARCHICAL RELATIVE POLICY OPTIMIZATION

We have aligned the Mas construction process with the RL by explicitly formulating its atomic policy actions in above discussion. In this subsection, we will introduce the reward mechanism that guides the framework toward learning an optimal Mas construction policy.

The evaluation of a Mas instance is inherently multi-dimensional, encompassing its performance quality, resource efficiency, and the rationality of its components. Prior studies have predominantly targeted only one or two of these dimensions, whereas RL enables a unified framework to pursue globally optimal Mas across all criteria. To this end, we propose a Hierarchical Relative Optimization (HRPO), which integrates group-relative advantages and step-wise action rewards.

**Group-relative advantage.** Balancing accuracy and efficiency is the core principle of constructed Mas. We introduce an intra-group advantage comparison mechanism to achieve this goal. By comparing relative advantages among instances, this mechanism generates preference signals that drive the policy network to pursue optimal objectives while minimizing resource consumption. Specifically, given the initial state $s_0$, we first sample a group of Mas instances based on the old policy $\pi_{old}$,

denoted as $\mathtt{G} = \{\mathtt{M}_1, \mathtt{M}_2, \ldots, \mathtt{M}_L\}$. Subsequently, instance $\mathtt{M}_i$ is evaluated in terms of both accuracy and resource efficiency in answering the same query $Q$. The reward function $r_{\mathtt{G}}(\cdot)$ is designed as,

$$r_{\mathtt{G}}(\mathtt{M}_i) = \begin{cases} 1 - \beta \cdot Tokens, & \mathtt{M}_i(Q) = \mathbf{Y} \\ -1, & \mathtt{M}_i(Q) \neq \mathbf{Y} \end{cases} \tag{4}$$

where $\mathbf{Y}$ is the ground-truth of query $Q$ and $\beta$ is a hyper-parameter to ensure $\beta \cdot Tokens \in [0, 1]$. Besides, $Tokens$ refers to the token usage ( the sum of prompt and completion tokens) by $\mathtt{M}_i$ in answering query $Q$. By implementing reward evaluation on each instance, we can collect the global rewards for the group as $R_{\mathtt{G}} = \{r_{\mathtt{G}}(\mathtt{M}_1), r_{\mathtt{G}}(\mathtt{M}_2), \ldots, r_{\mathtt{G}}(\mathtt{M}_L)\}$. In order to quantify the policy preferences through comparison, the normalized relative advantage of the $\mathtt{M}_i$ is computed as $A_{\mathtt{G}}(i) = \frac{r_{\mathtt{G}}(\mathtt{M}_i) - \bar{r}_{\mathtt{G}}}{\sigma_{r_{\mathtt{G}}}}$, where $\bar{r} = \mathrm{Mean}(R_{\mathtt{G}})$ and $\sigma_r = \mathrm{Var}(R_{\mathtt{G}})$. Therefore, $A_{\mathtt{G}}$ distills the strengths and weaknesses of each $\mathtt{M}$, which can effectively guide the policy network to favor superior patterns during training.

**Action-wise absolute reward.** Above relative advantage comparison mechanism can guarantee the performance and efficiency of the Mas, but fail to capture the rationality of its internal structure. To this end, we introduce an action-wise absolute reward to explicitly guide the rationality of internal structural design. Early-added agents, which may focus on task decomposition rather than delivering accurate answers, always initially show poor performance. These agents also play a pivotal role in structuring the collaborative process and enabling downstream success. Therefore, it is essential to protect and encourage these early-added agents to ensure the Mas fosters reasonable individual collaboration and gradual performance refinement. We introduce an exemption time $\mathcal{T}_E$ to safeguard early-stage exploration, where the actions taken before $\mathcal{T}_E$ are exempt from penalties, even if they fail to reach the correct solution. Based on this setting, we define the action-wise reward function in as follows:

$$r_{\mathtt{M}_i}(\mathtt{a}_t) = \begin{cases} -1, & \text{if } \mathcal{O}_{t-1} = \mathbf{Y}, \mathcal{O}_t \neq \mathbf{Y} \\ 1, & \text{if } \mathcal{O}_{t-1} \neq \mathbf{Y}, \mathcal{O}_t = \mathbf{Y} \\ e^{-t}, & \text{if } \mathcal{O}_t = \mathcal{O}_{t-1} = \mathbf{Y} \\ 0, & \text{if } t \leq \mathcal{T}_E, \mathcal{O}_t = \mathcal{O}_{t-1} \neq \mathbf{Y} \\ -\alpha \cdot (t - \mathcal{T}_E), & \text{if } t > \mathcal{T}_E, \mathcal{O}_t = \mathcal{O}_{t-1} \neq \mathbf{Y} \end{cases} \tag{5}$$

where $\alpha$ is a hyper-parameter to ensure $-\alpha \cdot (t - \mathcal{T}_E) \in [-1, 0]$, and $\mathcal{O}_{t-1}$ represents the intermediate output produced by the constructed Mas after executing action $\mathtt{a}_t$. This reward function evaluates the $t$-th action $\mathtt{a}_t$ taken during the construction of $\mathtt{M}_i$, following the principles outlined below.

- $\mathcal{O}_{t-1} = \mathbf{Y}, \mathcal{O}_t \neq \mathbf{Y}$. This scenario represents the worst case, where the current action $\mathtt{a}_t$ disrupts an already correct Mas. Therefore, it should be assigned the maximum penalty, even if it occurs before the exemption time.

- $\mathcal{O}_{t-1} \neq \mathbf{Y}, \mathcal{O}_t = \mathbf{Y}$. This represents the best-case scenario, indicating that the policy network has successfully captured the correct answering path. To this end, it is assigned the maximum reward when this occurs.

- $\mathcal{O}_t = \mathcal{O}_{t-1} = \mathbf{Y}$. This indicates that consistently correct answers are commendable. However, as the number of exploration steps increases, the reward should decay toward zero.

- $t \leq \mathcal{T}_E, \mathcal{O}_t = \mathcal{O}_{t-1} \neq \mathbf{Y}$. This case indicates that, before the exemption time, the current action $\mathtt{a}_t$ neither improves the previous incorrect outcome. This action is neutral and thus free of penalty.

- $t > \mathcal{T}_E, \mathcal{O}_t = \mathcal{O}_{t-1} \neq \mathbf{Y}$. The action $\mathtt{a}_t$ fails to bring about any changes in performance after the exemption time. While it does not worsen the result, it is still discouraged. This case may reflect an exploration failure of the policy network. Therefore, a significant penalty $-\alpha \cdot (t - \mathcal{T}_E) \in [-1, 0]$ increasing with $t$ is assigned to this action.

We have quantified the reward in Mas construction from both group-relative preference and action-level reward perspectives. The combination of these hierarchical rewards forms a composite action reward signal that collaboratively guide the policy function to design Mas with strong performance, high efficiency, and reasonable components. Building on this hierarchical reward design, the final action advantage $\hat{A}_i(\mathtt{a}_t)$ for each action $\mathtt{a}_t$ in $\mathtt{M}_i$ is formulated as,

$$\hat{A}_i(\mathtt{a}_t) = A_{\mathtt{G}}(i) + \sum_{T=t}^{|\mathtt{M}_i|} \gamma^{T-t} r_{\mathtt{M}_i}(\mathtt{a}_T). \tag{6}$$

Table 1: Performance comparison with single agent execution methods, hand-craft multi-agent systems, agentic workflows, and autonomous mutli-agent systems. The execution LLM is consistently set as gpt-4o-mini for all baselines. We report the average performance across five independent runs.

| Methods | GSM8K | MATH | MMLU | GPQA | MBPP | HumanEval | Average |
|---|---|---|---|---|---|---|---|
| IO OpenAI (2024) | 87.37 | 46.32 | 81.53 | 39.21 | 71.62 | 87.21 | 68.88 |
| CoT Wei et al. (2022) | $86.85_{\downarrow 0.52}$ | $45.83_{\downarrow 0.49}$ | $81.92_{\uparrow 0.39}$ | $39.20_{\downarrow 0.01}$ | $71.21_{\downarrow 0.41}$ | $88.39_{\uparrow 1.18}$ | 68.90 |
| SC (CoT×5) Wang et al. (2022) | $87.86_{\uparrow 0.49}$ | $47.79_{\uparrow 1.47}$ | $80.65_{\downarrow 0.88}$ | $38.98_{\downarrow 0.23}$ | $72.87_{\uparrow 1.25}$ | $88.37_{\uparrow 1.16}$ | 69.42 |
| MultiPersona Wang et al. (2023) | $87.12_{\downarrow 0.25}$ | $43.97_{\downarrow 2.35}$ | $81.03_{\downarrow 0.50}$ | $40.09_{\uparrow 0.88}$ | $72.18_{\uparrow 0.56}$ | $87.54_{\uparrow 0.33}$ | 68.66 |
| LLM-Debate Du et al. (2023) | $88.52_{\uparrow 1.15}$ | $47.33_{\uparrow 1.01}$ | $82.44_{\uparrow 0.91}$ | $39.57_{\uparrow 0.36}$ | $69.82_{\downarrow 1.80}$ | $88.07_{\uparrow 0.86}$ | 69.29 |
| DyLAN Liu et al. (2023) | $89.21_{\uparrow 1.84}$ | $48.19_{\uparrow 1.87}$ | $81.90_{\uparrow 0.37}$ | $40.54_{\uparrow 1.33}$ | $76.50_{\uparrow 4.88}$ | $86.98_{\downarrow 0.23}$ | 70.55 |
| GPTSwarm Zhuge et al. (2024) | $88.34_{\uparrow 0.97}$ | $48.31_{\uparrow 1.99}$ | $81.49_{\downarrow 0.04}$ | $42.41_{\uparrow 3.20}$ | $77.34_{\uparrow 5.72}$ | $88.29_{\uparrow 1.08}$ | 71.03 |
| ADAS Hu et al. (2024) | $85.72_{\downarrow 1.65}$ | $41.70_{\downarrow 4.62}$ | $80.61_{\downarrow 0.92}$ | $39.80_{\uparrow 0.59}$ | $68.00_{\downarrow 3.62}$ | $83.79_{\downarrow 3.42}$ | 66.60 |
| AFlow Zhang et al. (2024b) | $90.60_{\uparrow 3.23}$ | $50.63_{\uparrow 4.31}$ | $81.93_{\uparrow 0.40}$ | $44.23_{\uparrow 5.02}$ | $80.94_{\uparrow 9.32}$ | $89.27_{\uparrow 2.06}$ | 72.94 |
| AutoAgents Chen et al. (2023) | $87.36_{\downarrow 0.01}$ | $43.94_{\downarrow 2.38}$ | $82.00_{\uparrow 0.47}$ | $42.57_{\uparrow 3.36}$ | $71.11_{\downarrow 0.51}$ | $86.95_{\downarrow 0.26}$ | 68.99 |
| MAS-GPT Ye et al. (2025) | $91.36_{\uparrow 3.99}$ | $52.11_{\uparrow 5.79}$ | $82.09_{\uparrow 0.56}$ | $44.91_{\uparrow 5.70}$ | $80.19_{\uparrow 8.57}$ | $87.76_{\uparrow 0.55}$ | 73.07 |
| G-Designer Zhang et al. (2024a) | $91.27_{\uparrow 3.90}$ | $50.03_{\uparrow 3.71}$ | $81.44_{\uparrow 0.09}$ | $42.02_{\uparrow 2.81}$ | $80.10_{\uparrow 8.48}$ | $87.32_{\uparrow 0.11}$ | 72.03 |
| MaAS Zhang et al. (2025) | $91.76_{\uparrow 4.39}$ | $51.71_{\uparrow 4.40}$ | $83.17_{\uparrow 1.64}$ | $44.39_{\uparrow 5.18}$ | $80.21_{\uparrow 8.59}$ | $90.09_{\uparrow 2.88}$ | 73.56 |
| **MasHost** (Ours) | $93.23_{\uparrow 5.86}$ | $52.42_{\uparrow 6.10}$ | $83.40_{\uparrow 1.87}$ | $45.19_{\uparrow 5.98}$ | $80.97_{\uparrow 9.35}$ | $89.96_{\uparrow 2.75}$ | **74.20** |

The learning objective of our `MasHost` based on HPRO policy is formalized by,

$$\mathcal{J}_{\mathrm{HRPO}}(\theta, \phi) = \frac{1}{L} \sum_{i=1}^{L} \frac{1}{|\mathtt{M}_i|} \sum_{t=1}^{|\mathtt{M}_i|} \left\{ \min \left[ w(\theta, \phi) \cdot \hat{A}_{i,t}, \mathrm{clip}(w(\theta, \phi), 1 - \varepsilon, 1 + \varepsilon) \cdot \hat{A}_{i,t} \right] \right\},$$

$$w(\theta, \phi) = \frac{\pi_\theta(\mathtt{M}_{i,t}|\mathbf{q}, \mathrm{MESSAGE}(\mathtt{M}_{i,t-1}))}{\pi_{\theta_{old}}(\mathtt{M}_{i,t}|\mathbf{q}, \mathrm{MESSAGE}(\mathtt{M}_{i,t-1}))} \frac{\prod_{\mathtt{R}_j \in \mathtt{M}_{i,t-1}} \pi_\phi(e_{i,j}|\mathbf{q}, \mathtt{M}_{i,t-1}, \mathrm{MESSAGE}(\mathtt{R}_j))}{\prod_{\mathtt{R}_j \in \mathtt{M}_{i,t-1}} \pi_{\phi_{old}}(e_{i,j}|\mathbf{q}, \mathtt{M}_{i,t-1}, \mathrm{MESSAGE}(\mathtt{R}_j))}, \quad (7)$$

where $\pi_\theta$ and $\pi_\phi$ denote the current policy models, and $\pi_{\theta_{old}}$ and $\pi_{\phi_{old}}$ represent the corresponding old policy models. $\varepsilon$ is a clipping-related hyper-parameter introduced in PPO Schulman et al. (2017) for stabilizing training. Similarly, $w(\theta, \phi)$ denotes the importance sampling ratio, also introduced in PPO, which serves to constrain excessive policy updates by adjusting the weight of sampled Mas.

## 5 EXPERIMENTS

### 5.1 EXPERIMENTAL SETUP

**Datasets.** We evaluate `MasHost` on six public benchmarks, including (1) math reasoning: GSM8K Cobbe et al. (2021), MATH Hendrycks et al. (2021); (2) question-answering: GPQA Rein et al. (2024), MMLU Hendrycks et al. (2020); (3) code generation: HumanEval Chen et al. (2021), MBPP Austin et al. (2021). Detailed dataset descriptions and divisions are provided in Appendix. C.1.

**Baselines.** We compare mutli-agent systems constructed by `MasHost` against various types of baselines, including (1) single agent execution methods: IO OpenAI (2024), Chain-of-Thought (CoT) Wei et al. (2022), CoT SC (5-shot) Wang et al. (2022); (2) hand-craft multiagent systems: MultiPersona Wang et al. (2023), LLM-Debate Du et al. (2023), DyLAN Liu et al. (2023); (3) agentic workflows: GPTSwarm Zhuge et al. (2024), ADAS Hu et al. (2024), AFlow Zhang et al. (2024b); (4) autonomous mutli-agent systems: AutoAgents Chen et al. (2023), MAS-GPT Ye et al. (2025), G-Designer Zhang et al. (2024a), MaAS Zhang et al. (2025).

**Implementation Details.** Following the experimental settings adopted by most baselines Zhang et al. (2024b; 2025), we select GPT-4o-mini-0718 OpenAI (2024) as the LLM executor, which is accessed via APIs. Besides, we set the temperature to 0 for the executor. We implement our `MasHost` on a server equiped with an NVIDIA A100-SXM4-80GB GPU. The code is available at https://anonymous.4open.science/r/MasHost-81BF.

**Metrics.** For GSM8K, MATH, GPQA and MMLU, we report the Accuracy (%) as the metric. For HumanEval and MBPP, we report the Pass@1 (%) to assess code accuracy.

## 5.2 PERFORMANCE COMPARISON

As shown in Tab. 1, our proposed `MasHost` consistently achieves the best performance among all compared methods. Compared to the existing state-of-the-art, our `MasHost` achieves an absolute performance improvement of up to 1.47 % on the GSM8k, highlighting its superiority over existing methods. Furthermore, we also focus on the samples where `MasHost` failed to provide correct answers to further investigate its robustness. We categorize the samples with incorrect answers into five types: (1) global failure due to *Incorrect Role Selection* (*IRS*), (2) target omission caused by *Task Forgetting* (*TF*), (3) incomplete answers caused by *Premature Termination* (*PT*), (4) *Incorrect Verification* (*IV*), and (5) correct reasoning with *Slight Deviations* in the final result (*SD*). As shown in Fig. 3*(left)*, we observe that the erroneous samples are primarily concentrated in two categories: IV and SD. This indicates that `MasHost` is able to identify the correct direction for answering but fails to produce the correct solution due to the complexity and difficulty of the questions. This demonstrates the potential of `MasHost` in tackling complex problems and highlights its robustness.

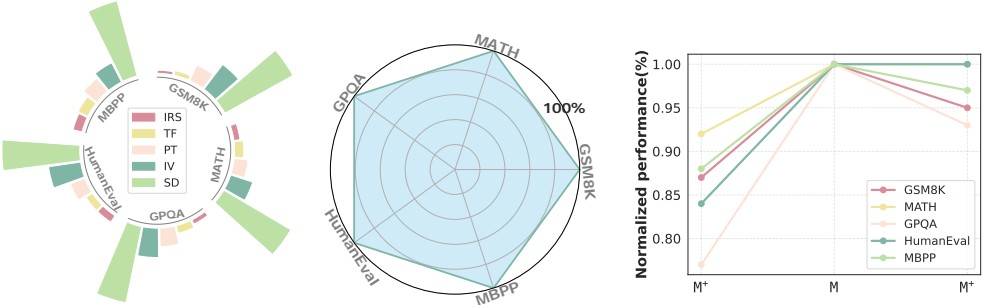

Figure 3: *(left)* Robustness of `MasHost`. *(middle)* The similarity between the roles and the queries type. *(right)* Rationality of Constructed Mas.

## 5.3 COST-EFFICIENT ANALYSIS

As shown in Tab. 2, we present the average cost required to answer each query in the test phase, using GPT-4o-mini as execution LLM. The cost efficiency of our `MasHost` is highly competitive. Actually, we have incorporated the following design strategies into our framework to reduce costs. (1) The inter-group advantage in HRPO takes cost consumption into account and quantifies the associated loss. (2) The global message pool prevents redundant invocations of the same role. Therefore, we conclude that our `MasHost` provides performance improvements while maintaining cost efficiency.

Table 2: Efficiency comparison on the MATH Benchmark. Best results are in **bold**.

| Methods | # Prompt Tokens | # Completion Tokens | Cost ($\times 10^{-3}$ *USD*) |
|---|---|---|---|
| GPTSwarm | 6,322 | **3,379** | 2.976 |
| AFlow | 4,938 | 3,943 | 3.106 |
| MaAS | 5,273 | 3,749 | 3.040 |
| **MasHost** | **3,630** | 3,698 | **2.763** |

## 5.4 RATIONALITY DISCUSSION

We assess the rationality of the multi-agent system built by `MasHost` from two aspects: (1) the role rationality and (2) the structure rationality.

**Rationality of role assignment.** Given the full-scale role space search in our work, ensuring the rationality of role selection is essential for tackling complex real-world queries. We design a correlation matching strategy to verify whether each role in the constructed Mas is relevant to the given query. As shown in Fig. 3*(middle)*, we observe a perfect correlation (i.e., 100%) between the assigned roles and query types across all datasets. This demonstrates that even under full-space role search, the multi-agent system constructed by `MasHost` maintains explainable rationality.

**Rationality of Mas structure.** We evaluate the rationality of our Mas structure in terms of redundancy and oversimplification. Let M denote the multi-agent system generated by `MasHost`, where

removing one agent yields $M^-$ and adding one task-related agent results in $M^+$. We sample 100 instances from each of the GSM8K and HumanEval datasets to compare the performance of $M$, $M^-$, and $M^+$, thereby verifying the rationality of the constructed MAS. As shown in Fig 3(*fight*), we observe that, compared to $M$, the performance of $M^-$ exhibits a significant drop, while the performance of $M^+$ show a slight performance degradation. The decline in performance resulting from the addition of agents is primarily due to incorrect post-processing, which can corrupt previously accurate information. This indicates that the $M$ constructed by `MasHost` achieves an efficient, accurate, and reasonable multi-agent system.

## 5.5 ABLATION STUDY

We conduct ablation studies to explore the effectiveness of each component of `MasHost`. Specifically, we analyze the respective impacts of three core components: the joint probabilistic space sampling mechanism (*JPSS*), hierarchical relative policy optimization (*HRPO*), and the design of exemption time (*ET*). To this end, we design three variants based on `MasHost`, `MasHost` *w.o.* *JPSS*, `MasHost` *w.o.* *HRPO*, and `MasHost` *w.o.* *ET*. Tab. 3 shows that the performance drops significantly when any of the three core components is removed. Among them, `MasHost` *w.o.* *HRPO* exhibits the most

Table 3: Ablation study of `MasHost`. *Cost* refers to the relative proportion of total token consumption during the training, with `MasHost` normalized to 1.00.

| Dataset | HumanEval | | GSM8K | |
|---|---|---|---|---|
| | Perf. | Cost | Perf. | Cost |
| `MasHost` | 89.96 | 1.00 | 93.23 | 1.00 |
| `MasHost` *w.o. JPSS* | 88.07 | 1.10 | 91.53 | 1.02 |
| `MasHost` *w.o. HRPO* | 87.22 | 1.43 | 90.64 | 1.73 |
| `MasHost` *w.o. ET* | 88.93 | 0.92 | 91.17 | 0.96 |

significant performance drop, indicating that this component has the greatest impact on performance. Although `MasHost` *w.o.* *ET* has a relatively smaller effect on performance, the resulting multi-agent systems often converge to a smaller scale. In this case, many of the resulting structures lack rationality and fail to handle complex tasks effectively.

## 5.6 SENSITIVITY ANALYSIS

We investigate the sensitivity of training rounds $n_r$, and exemption time $\mathcal{T}_E$. As shown in Fig. 4, we present the performance fluctuations under different hyper-parameter settings on GSM8K and HumanEval. Although performance improves with larger $n_r$, the marginal gains diminish when $n_r > 4$. Therefore, we fix $n_r = 4$ to achieve a trade-off between performance and cost. We observe that the performance converges once the exemption time $\mathcal{T}_E > 3$. Given that the value of $\mathcal{T}_E$ is proportional to the cost consumption, we set $\mathcal{T}_E = 3$. Moreover, the hyper-parameters $\alpha$, $\beta$, $\gamma$, and $\varepsilon$ are discussed in detail in Appendix C.2.

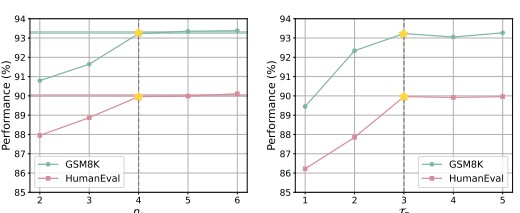

Figure 4: The sensitivity of training rounds $n_r$, and exemption time $\mathcal{T}_E$.

## 6 CONCLUSION

In this work, we propose `MasHost`, a novel reinforcement learning-based framework that enables the fully autonomous construction of query-specific Multi-agent system. By introducing a joint probabilistic sampling mechanism and a novel Hierarchical Relative Policy Optimization strategy, `MasHost` enables end-to-end autonomous design of multi-agent systems with enhanced adaptability, rationality, and performance. Our approach enables scalable, efficient, and interpretable construction of autonomous Mas. **Limitations.** A key application of LLM-based multi-agent systems lies in tool usage for solving real-world problems, which is beyond the scope of our current work. To enable this capability, it is essential to disentangle the relationship between multi-agent interaction patterns and tool utilization. In future work, we plan to explore methods for effectively decoupling multi-agent interaction modes from tool usage, enabling more flexible and adaptive integration of external tools.

ETHICS STATEMENT

Our `MasHost` is an LLM-based multi-agent system designed to address the limitations and performance challenges of a single LLM. As it does not involve human subjects, nor raise concerns related to discrimination, bias, or fairness, this work involves no potential ethical hazards.

REPRODUCIBILITY STATEMENT

No private datasets are used in this paper, and the complete source code has been released via anonymous links. Detailed descriptions of the experimental settings are provided in the maintext and the code repository to ensure reproducibility.

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

## THE USE OF LARGE LANGUAGE MODELS

This paper makes use of large language models in two respects. Firstly, `MasHost` is an LLM-based multi-agent system, where the decision-making component of each agent is instantiated by an LLM. Secondly, during the presentation of this manuscript, large language models are used solely for language polishing, without activating any content generation capabilities.

## A  AUTONOMY AND RATIONALITY GUARANTEE

We guarantee the autonomous capability of `MasHost` to construct multiple agents from two complementary perspectives. Our HRPO-based graph growth mechanism can generate arbitrary directed graphs, while our role sampling strategy, in contrast to prior methods restricted to task-specific role pools, operates over the entire role space.

**Autonomy in graph construction.** From a graph-theoretic perspective, we argue that the design space explored by `MasHost` is equivalent to the entire set of directed graphs over a given node set. Specifically, by modeling node role assignments and edge connectivity as joint probabilistic variables, our framework ensures the representational completeness of all possible Mas interaction topologies without structural bias or limitation. This guarantee implies that `MasHost` can generate any feasible directed graph configuration, thus achieving full autonomy in graph construction.

**Autonomy of role selection.** The autonomous capability of role selection is largely overlooked in existing works, which typically preset a task-specific role pool and select agent roles within this limited space. In this work, we focus on enabling autonomous role selection by sampling from the entire role space without human-imposed restrictions. This approach not only enhances the flexibility and generality of the system but also allows for emergent agent behaviors that are better aligned with dynamic task demands. To address the associated optimization challenges arising from the high-dimensional and combinatorial nature of the full role space, we introduce a joint probabilistic modeling framework that guides role sampling in a stable and differentiable manner.

---

**Algorithm 1** MASHOST: RL-based Multi-Agent System Construction

---

**Require:** Query $Q$, full-scale role pool $\mathcal{R}$
**Ensure:** Multi-agent System Graph M
 1: Initialize policy networks $\pi_\theta$ (node-level), $\pi_\phi$ (edge-level)
 2: Initialize empty MAS graph $\mathtt{M} \leftarrow \emptyset$, $s_0 = \{Q\}$
 3: **while** not TERMINATED(M) **do**
 4:     Observe current state $s_t = \{Q, \mathtt{M}, \text{MESSAGE}(\mathtt{M})\}$
 5:     Sample 4 cases to construct a relative group $\mathtt{G} = \{\mathtt{M}_1, \mathtt{M}_2, \mathtt{M}_3, \mathtt{M}_4\} \sim \pi$
 6:     Sample action $a_n \sim \pi_\theta(a_n \mid s_t)$            ▷ Node-level action
 7:     **if** $a_n =$ EXIT **then**
 8:         **break**
 9:     **else if** $a_n =$ DELETE **then**
10:         Remove last-added agent from M
11:     **else**
12:         Add agent $v$ with role $a_n$ to M
13:         Sample edge distribution $P_e \leftarrow \pi_\phi(a_e \mid s_t, a_n)$       ▷ Edge-level action
14:         Sample connections $a_e \sim p(a_n) \cdot P_e$      ▷ Joint distribution sampling
15:         Add edges $a_e$ to M
16:     **end if**
17:     Compute group-relative preference $A_\mathtt{G}(i)$ and action-level reward $r_{\mathtt{M}_i}(\mathtt{a}_T)$
18:     Compute advantage $\hat{A}_i(\mathtt{a}_t) = A_\mathtt{G}(i) + \sum\limits_{T=t}^{|\mathtt{M}_i|} \gamma^{T-t} r_{\mathtt{M}_i}(\mathtt{a}_T)$
19:     Update $\pi_\theta, \pi_\phi$ via HRPO objective $\mathcal{J}_{\text{HRPO}}(\theta, \phi)$
20: **end while**
21: **return** M

---

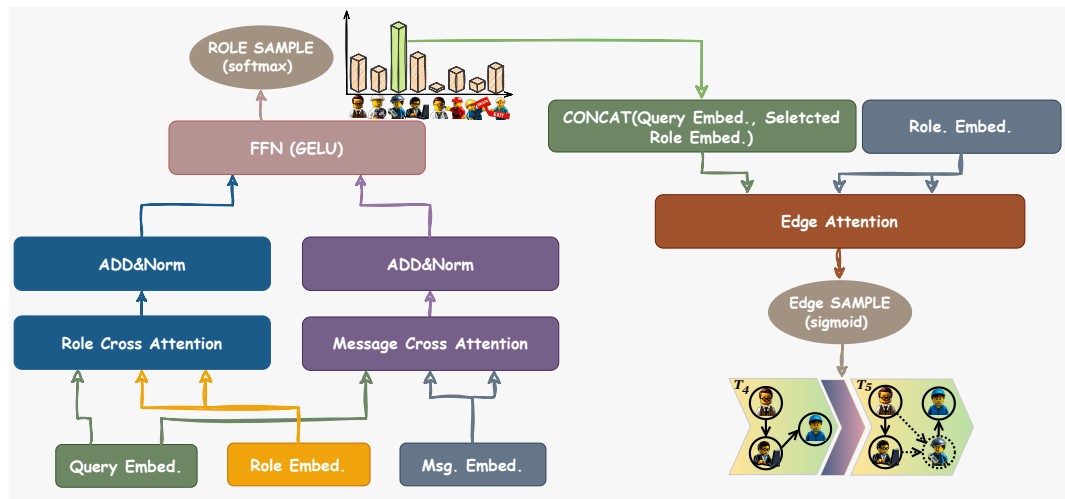

Figure 5: The network framework of our MasHost.

# B    VISUALIZATION RESULTS

To intuitively demonstrate the effectiveness of our `MasHost`, we visualize the constructed multi-agent system. As shown in Fig. 6 7 8 9, our `MasHost` yields agents with clearly distinguishable roles and behaviors, offering strong interpretability in both structure and decision-making. The visualized trajectories and interactions not only align well with real-world patterns but also reflect the model's superior performance in terms of coordination and task success. These visualizations compellingly demonstrate that our approach achieves a strong balance between interpretability and performance.

# C    IMPLEMENTATION DETAILS

## C.1    DATASET DESCRIPTIONS AND DIVISIONS

We partially follow the data partitioning scheme of Zhang et al. (2024b), while adapting the split to align with the characteristics of our framework, as shown in Fig. 4

Table 4: Dataset descriptions and partitions.

| Dataset | Sample size | Sampling Strategy | Training set | Test set |
|---|---|---|---|---|
| GSM8K | 8,500 | Full Samples | 200 | ~6,800 |
| MATH | ~12,500 | Select 617 | 123 | 494 |
| MBPP | 974 | Full Samples | 200 | ~779 |
| HumanEval | 164 | Full Samples | 33 | 131 |
| GPQA | 448 | Full Samples | 200 | 248 |
| MMLU | ~15,908 | Select 1,000 | 200 | 800 |

## C.2    HYPER-PARAMETERS SETTINGS

The hyper-parameters $\alpha$, $\beta$, $\gamma$, and $\varepsilon$ play a critical balancing role in our framework, mediating trade-offs between reward shaping and learning objectives to ensure stable and effective policy optimization. In this section, we elaborate on their functionality and the specific settings adopted in our implementation.

| Query （GSM8K） | Multi-agent system |
|---|---|
| Steve is 60 years old. His wife is 4 years older than him. Their son is currently half as old as his mom and their son's wife is 3 years younger than her husband. How old is Steve's son's wife? | *Algebra solver*      *Summarizer* |

Figure 6: The Mas constructed on the GSM8K sample.

| Query （HumanEval） | Multi-agent system |
|---|---|
| Help me implement this function: def is_prime(n):   """Return true if a given number is prime, and false otherwise.     >>> is_prime(6)   False     >>> is_prime(101)   True     >>> is_prime(11)   True     >>> is_prime(13441)   True     >>> is_prime(61)   True     >>> is_prime(4)   False     >>> is_prime(1)   False     """ | *Python expert*      *Python expert* 
 *Algebra solver*      *Summarizer* |

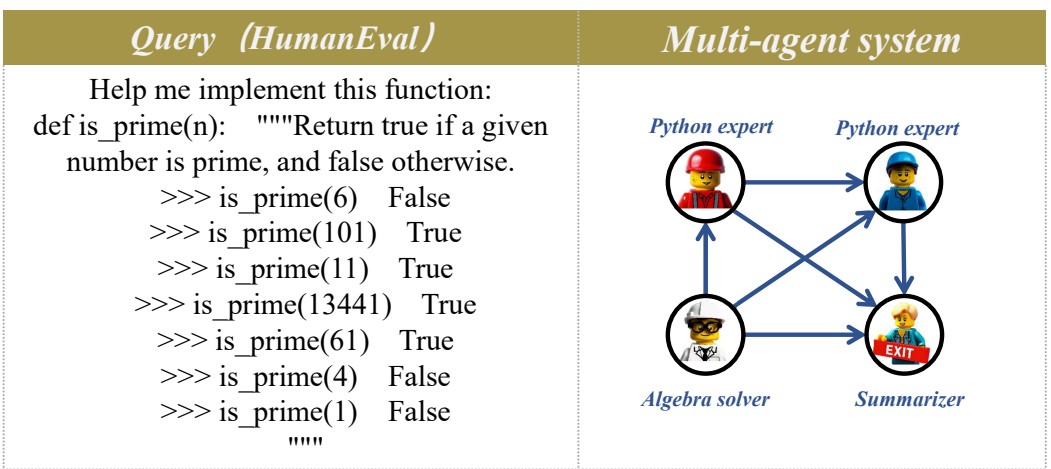

Figure 7: The Mas constructed on the HumanEval sample.

| Query （Math） | Multi-agent system |
|---|---|
| If $|x+5|-|3x-6|=0$, find the largest possible value of $x$. Express your answer as an improper fraction. | |

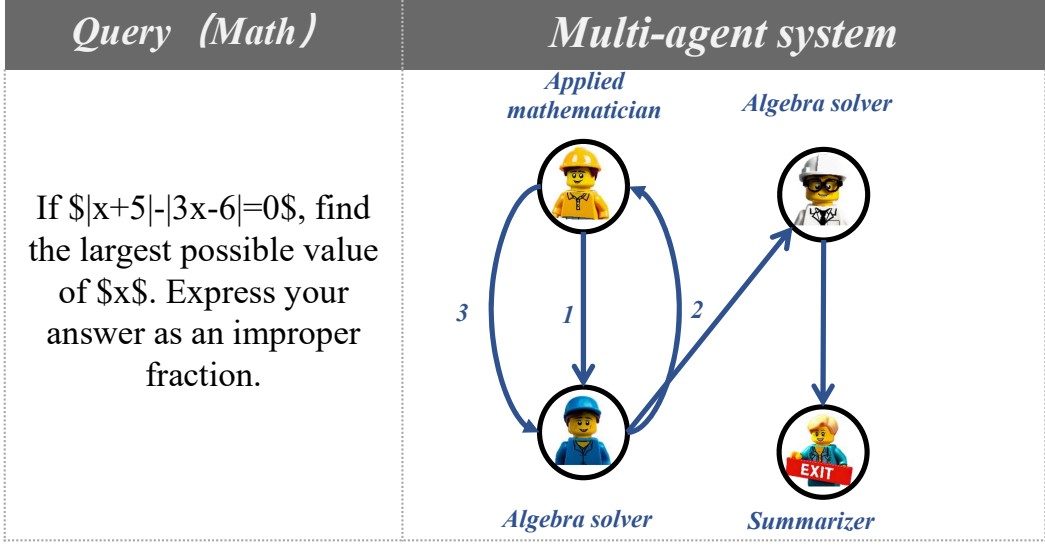

Figure 8: The Mas constructed on the MATH sample.

| Query （MBPP） | Multi-agent system |
|---|---|
| Write a function to find the peak element in the given array. Your code should pass these tests: ```python     assert find_peak([1, 3, 20, 4, 1, 0], 6) == 2     assert find_peak([2, 3, 4, 5, 6], 5) == 4     assert find_peak([8, 9, 11, 12, 14, 15], 6) == 5 ``` | 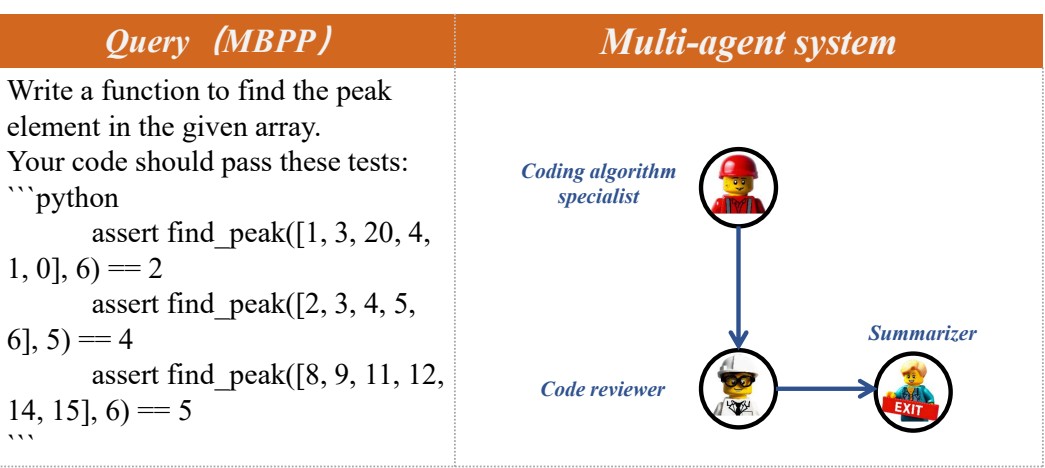 |

Figure 9: The Mas constructed on the MBPP sample.

| Query （MMLU） | Multi-agent system |
|---|---|
| Let A and B be sets, f: A -> B and g: B -> A be functions such that for all a \in A, g(f(a)) = a. Statement 1 \| The function f must necessarily be injective. Statement 2 \| The function f must necessarily be surjective. A: True, True B: False, False C: True, False D: False, True | 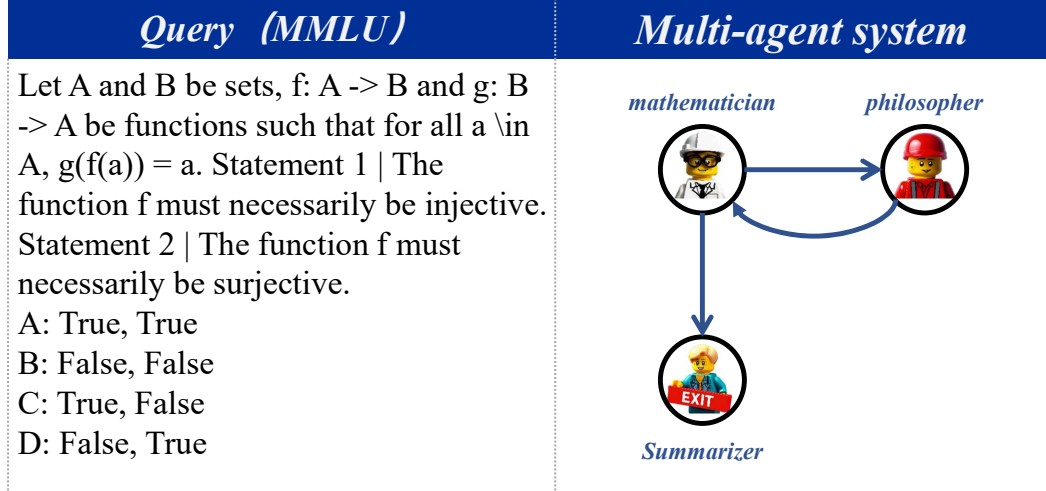 |

Figure 10: The Mas constructed on the MMLU sample.

- The $\alpha$ in Eq. 5 is set $0.1$ in the implementation. The $\alpha$ is a balancing hyper-parameter to ensure $-\alpha \cdot (t - \mathcal{T}_E) \in [-1, 0]$. Since the number of exploration steps typically does not exceed 10, the value $\alpha$ is empirically set to $0.1$.

- The hyperparameter $\beta$ in Eq. 4 is set to $0.0001$ for GSM8K and $0.00001$ for the other datasets. The $\beta$ is a balancing hyper-parameter to ensure $\beta \cdot Tokens \in [0, 1]$. The difference setting mainly stems from that the number of tokens consumed per answer in GSM8K ranges from $100$ to $1,000$, whereas in the other datasets, it typically ranges from $1,000$ to $10,000$.

- The parameter $\gamma$ in Eq. 6 is set to $0.9$ in our implementation, following common configurations adopted in reinforcement learning practices. The discount factor $\gamma$ controls the temporal weighting of future rewards, enabling the agent to balance short-term gains with long-term objectives.

- The parameter $\varepsilon$ in Eq. 7 is set to $0.1$ in our implementation, following the configuration used in the paper Shao et al. (2024); Schulman et al. (2017). The clipping threshold $\varepsilon$ constrains policy updates by limiting the change in the probability ratio, thus preventing overly aggressive updates that could destabilize trainingSchulman et al. (2017).

## C.3 ROLE PROMPTS

Our `MasHost` relies on a global role pool, which includes all known applicable roles. We provide specific role names along with corresponding prompts. Different from existing practices, they overlook the design of refuse conditions for agent. We highlight the specific function and identity of each role agent. This design is motivated by the aim of this work to enhance the rationality of Mas. The irrationality of previous methods lies in their tendency to allow the model to select a role completely unrelated to the question, yet still generate a valid response, as shown in Fig. 11. While this may seem acceptable for relatively simple problems, it hinders broader transfer and real-world applicability.

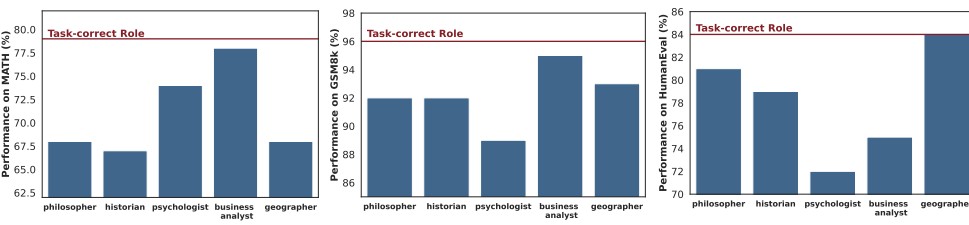

(a) Performance of unreasonable role assignment on MATH.

(b) Performance of unreasonable role assignment on GSM8k.

(c) Performance of unreasonable role assignment on HumanEval.

Figure 11: Roles associated with unrelated tasks are nevertheless able to answer the queries well.

---

**Molecular Biologist**

**Responsibilities:**

- Study structure and function of biomolecules (DNA, proteins, etc.)
- Analyze gene expression and regulation
- Investigate molecular mechanisms of cellular processes
- Develop techniques like PCR or CRISPR

**Assist Conditions:**

- General biology questions
- Related fields (e.g., Genetics, Biochemistry, Biotechnology)

---

## Cell Biologist

**Responsibilities:**

- Study cell structure, division, and metabolism
- Investigate cell signaling and communication
- Analyze organelle functions (e.g., mitochondria, nucleus)
- Research cell responses to environmental changes

**Assist Conditions:**

- General biology questions
- Related fields (e.g., Molecular Biology, Immunology, Cancer Research)

## Geneticist

**Responsibilities:**

- Study inheritance patterns and genetic variation
- Analyze DNA sequencing data
- Investigate genetic disorders
- Develop genetic engineering tools

**Assist Conditions:**

- General biology questions
- Related fields (e.g., Genomics, Evolutionary Biology, Medicine)

## Botanist

**Responsibilities:**

- Study plant physiology and taxonomy
- Investigate plant-environment interactions
- Research photosynthesis and plant hormones
- Explore plant biodiversity and conservation

**Assist Conditions:**

- General biology questions
- Related fields (e.g., Ecology, Agriculture, Forestry)

## Biomedical Scientist

**Responsibilities:**

- Research disease mechanisms (e.g., cancer, infections)
- Develop diagnostic tools and therapies
- Study drug interactions and pharmacokinetics
- Investigate immune system responses

**Assist Conditions:**

- General biology questions
- Related fields (e.g., Pharmacology, Immunology, Clinical Research)

### Inorganic Chemist

**Responsibilities:**
- Study the structure and properties of inorganic compounds
- Investigate catalysis and reaction mechanisms in inorganic systems
- Develop new materials
- Analyze metal-ligand interactions in coordination chemistry
- Explore bioinorganic chemistry

**Assist Conditions:**
- General chemistry questions
- Related fields (e.g., Materials Science, Geochemistry, Industrial Catalysis)

### Organic Chemist

**Responsibilities:**
- Study the synthesis, structure, and reactivity of organic compounds
- Develop new synthetic methodologies
- Investigate reaction mechanisms
- Design pharmaceuticals, agrochemicals, or polymers
- Analyze spectroscopic data (NMR, IR, MS) for structure elucidation

**Assist Conditions:**
- General chemistry questions
- Related fields (e.g., Medicinal Chemistry, Polymer Science, Petrochemistry)

### Analytical Chemist

**Responsibilities:**
- Develop and optimize analytical techniques
- Perform qualitative and quantitative analysis of chemical samples
- Validate methods for quality control
- Interpret data from instruments
- Ensure compliance with regulatory standards

**Assist Conditions:**
- General chemistry questions
- Related fields (e.g., Forensic Science, Environmental Monitoring, Food Safety)

### Materials Chemist

**Responsibilities:**

- Design and synthesize novel materials
- Study structure-property relationships in materials
- Develop functional materials for energy storage
- Investigate smart materials

**Assist Conditions:**

- General chemistry questions
- Related fields (e.g., Nanotechnology, Electronics, Energy Science, Biomedical Engineering)

### Theoretical Chemist

**Responsibilities:**

- Develop computational models to predict molecular properties and reactions
- Apply quantum mechanics (e.g., DFT, ab initio methods) to chemical systems
- Simulate molecular dynamics and statistical mechanics
- Analyze chemical bonding and electronic structure
- Collaborate with experimentalists to interpret data and guide research

**Assist Conditions:**

- General chemistry questions
- Related fields (e.g., Computational Chemistry, Drug Design, Catalysis, Astrophysics)

### Code Reviewer

**Responsibilities:**

- Analyze code style compliance
- Identify potential bugs and security vulnerabilities
- Suggest performance optimizations
- Evaluate code readability and maintainability
- Check boundary conditions and exception handling

**Reject Conditions:**

- user mentioned that currently no cooperators available.
- Or user gives cooperators, but their messages are not related to code.

## Code Reviewer

**Responsibilities:**

- Analyze code style compliance
- Identify potential bugs and security vulnerabilities
- Suggest performance optimizations
- Evaluate code readability and maintainability
- Check boundary conditions and exception handling

**Reject Conditions:**

- user mentioned that currently no cooperators available.
- Or user gives cooperators, but their messages are not related to code.

## Debug Assistant

**Responsibilities:**

- Parse error messages and stack traces
- Locate root causes in code
- Suggest debugging methods and tools
- Verify effectiveness of fixes
- Reproduce and isolate error scenarios

**Reject Conditions:**

- user mentioned that currently no cooperators available.
- Or user gives cooperators, but their messages are not related to code.

## Python Programmer

**Responsibilities:**

- Answer Python language feature questions
- Explain standard library and third-party package usage
- Guide Python best practices
- Analyze advanced features
- Compare differences between Python implementations

**Assist Conditions:**

- General mathematics or physics questions

## Coding Algorithm Specialist

**Responsibilities:**

- Design optimal algorithms for problems
- Analyze time and space complexity
- Suggest suitable data structures
- Compare different algorithmic approaches
- Explain algorithm design patterns

**Assist Conditions:**

- General mathematics or physics questions

## Performance Optimizer

**Responsibilities:**

- Identify performance bottlenecks
- Suggest low-level optimizations
- Analyze memory usage patterns
- Guide parallelization strategies
- Recommend profiling tools and techniques

**Reject Conditions:**

- user mentioned that currently no cooperators available.
- Or user gives cooperators, but their messages are not related to code.

## Algebra Solver

**Responsibilities:**

- Solve linear and nonlinear equations
- Perform matrix operations and linear algebra computations
- Factor and manipulate polynomial expressions
- Solve systems of equations
- Simplify algebraic expressions

**Assist Conditions:**

- General mathematics questions
- Related fields (e.g., number theory, geometry)

## Geometry Specialist

**Responsibilities:**

- Explain coordinate geometry concepts
- Analyze geometric transformations
- Compute areas, volumes and angles
- Guide vector geometry applications
- Solve trigonometric problems

**Assist Conditions:**

- General mathematics questions
- Related fields (e.g., Physics applications, Computer graphics, Architectural design)

## Applied Mathematician

**Responsibilities:**

- Bridge theoretical math and practical applications
- Solve mathematical modeling problems
- Explain numerical analysis methods
- Guide optimization problem solutions
- Analyze operations research problems

**Assist Conditions:**

- General mathematics questions
- Related fields (e.g., Engineering problems, Economic modeling, Scientific computing)

## Analytic Mathematician

**Responsibilities:**

- Study limits, continuity, and convergence in real and complex spaces
- Develop theories in calculus, measure theory, and functional analysis
- Solve differential equations and harmonic analysis problems
- Explore Fourier analysis and operator theory
- Investigate partial differential equations and their applications

**Assist Conditions:**

- General mathematics questions
- Related fields (e.g., Mathematical physics, Dynamical systems, Probability theory)

## Discrete Mathematician

**Responsibilities:**

- Study combinatorial structures and graph theory
- Solve problems in cryptography and coding theory
- Analyze discrete optimization and algorithmic complexity
- Explore logic, set theory, and discrete probability
- Investigate network science and computational geometry

**Assist Conditions:**

- General mathematics questions
- Related fields (e.g., Computer science, Cryptography, Operations research)

---

### Classical Physicist

**Responsibilities:**

- Study macroscopic physics (mechanics, thermodynamics, electromagnetism)
- Analyze motion and forces in Newtonian frameworks
- Model wave phenomena and fluid dynamics
- Explain classical field theories

**Assist Conditions:**

- General physics questions
- Related fields (e.g., Engineering mechanics, Acoustics, Thermodynamic systems)

---

### Particle Physicist

**Responsibilities:**

- Investigate fundamental particles and interactions
- Interpret data from colliders (e.g., LHC)
- Test predictions of the Standard Model
- Explore beyond-Standard-Model theories

**Assist Conditions:**

- General physics questions
- Related fields (e.g., Quantum field theory, Cosmology, Nuclear physics)

---

### Quantum Physicist

**Responsibilities:**

- Study quantum systems and entanglement
- Develop quantum computing/algorithms
- Analyze atomic/subatomic behavior
- Explain quantum measurement problems

**Assist Conditions:**

- General physics questions
- Related fields (e.g., Quantum chemistry, Nanotechnology, Quantum optics)

---

### Condensed Matter Physicist

**Responsibilities:**

- Research solid/liquid state properties
- Study superconductivity or topological materials
- Model phase transitions and collective phenomena
- Design novel materials (e.g., graphene)

**Assist Conditions:**

- General physics questions
- Related fields (e.g., Semiconductor physics, Materials science, Spintronics)

> **Relativistic Physicist**
>
> **Responsibilities:**
>
> - Analyze spacetime curvature (GR effects)
> - Model black hole/neutron star dynamics
> - Test Lorentz invariance and relativistic jets
> - Simulate gravitational wave sources
>
> **Assist Conditions:**
>
> - General physics questions
> - Related fields (e.g., Astrophysics, Cosmology, High-energy physics)

## D  LIMITATIONS AND FUTURE WORK

A key application of LLM-based multi-agent systems lies in tool usage for solving real-world problems, which is beyond the scope of our current work. To enable this capability, it is essential to disentangle the relationship between multi-agent interaction patterns and tool utilization.

In future work, we plan to explore methods for effectively decoupling multi-agent interaction modes from tool usage, enabling more flexible and adaptive integration of external tools. Instead of relying on static workflows, we aim to introduce a unified decision-making framework in which agents can autonomously determine when, how, and which tools to invoke based on dynamic task demands. This will empower the system to generalize more effectively across real-world scenarios, supporting more robust problem-solving. By aligning tool usage with emergent agent collaboration dynamics, we anticipate improved coordination efficiency and a deeper synergy between reasoning, communication, and execution.

## E  BROADER IMPACTS

Our proposed `MasHost` framework focuses on enabling fully autonomous and adaptive multi-agent system design to improve the efficiency and effectiveness of solving complex real-world problems. By automating the generation and coordination of many agents, `MasHost` aims to reduce human labor in address real-world tasks, thereby empowering users to achieve higher productivity and better resource utilization. Therefore, `MasHost` itself does not involve mechanisms that pose direct ethical or safety risks. Instead, it serves as a foundational tool designed to assist researchers and practitioners in building more capable multi-agent systems with improved rationality, performance, and cost-efficiency. Our frameworks goal is to support human-centered applications by augmenting human capabilities rather than replacing them.

