# OpenReview forum: "MasHost Builds It All: Autonomous Multi-Agent System Directed by Reinforcement Learning"
_ICLR.cc/2026/Conference — ICLR 2026 Conference Withdrawn Submission_

### Official Review · Reviewer_AWgu · 2025-10-22

**Soundness:** 2
**Presentation:** 3
**Contribution:** 2
**Rating:** 4
**Confidence:** 3

**Summary:**

This paper introduces MasHost, an ambitious framework that autonomously builds multi-agent systems (MAS) from scratch using reinforcement learning. The system learns both the role composition of agents and their communication topology via a **Joint Probabilistic Space Sampling (JPSS)** procedure, optimized with a **Hierarchical Relative Policy Optimization (HRPO)** objective. The authors aim to remove human design bias in MAS formation by allowing the system to self-discover the optimal collaboration structure. Experiments are conducted on multiple reasoning benchmarks (math problem solving, code synthesis, and QA), using GPT-4o-mini as the executor, and the results show strong improvements compared to handcrafted and previous autonomous MAS baselines. Ablations further attribute the gains primarily to HRPO and the proposed “exemption time” scheduling.

**Strengths:**

* The problem of automating MAS architecture design is both novel and challenging, fitting well within the emerging trend of self-organizing multi-agent systems.

* The proposed JPSS + HRPO combination presents a clean and unified training pipeline that bridges architecture search and RL optimization.

* The experiments are fairly comprehensive in terms of task variety, demonstrating potential generality.

* The qualitative examples provide a clear sense of how the MAS evolves and learns agent specializations over time.

* The writing is accessible, and the visuals (especially the system overview) are informative and well-labeled.

**Weaknesses:**

* The executor dependency is a serious concern: results rely heavily on GPT-4o-mini (with temperature = 0), and no experiments are provided using alternative language models. This limits reproducibility and generality.

* The technical details of HRPO are underdeveloped. It is unclear how the advantage function is computed when two interdependent stochastic policies (role and edge) are being optimized simultaneously. There is no discussion of gradient variance or credit assignment.

* The **rationality** and **exemption time** concepts are vaguely defined, and lack formal mathematical justification.

* The experimental setup omits compute-matching details (e.g., number of tokens, API cost, and wall-clock time) which makes comparison fairness uncertain.

* No evaluation is performed on embodied or tool-use tasks, which would strengthen the claim that MasHost can generalize beyond static reasoning.

**Questions:**

1. How are the advantages for the role and edge policies estimated and combined during the HRPO updates?

2. Can you show results when using different executors (e.g., Claude, Gemini, or open-weight models) to demonstrate model-agnostic performance?

3. Could you include compute- and token-matched comparisons to control for inference variance?

4. How is **rationality** defined quantitatively, and how sensitive is performance to this component?

---

### Official Review · Reviewer_L5Te · 2025-10-27

**Soundness:** 2
**Presentation:** 2
**Contribution:** 2
**Rating:** 4
**Confidence:** 3

**Summary:**

This paper proposes MasHost, a reinforcement learning based framework that can automatically design a multi agent system, both the agents and how they talk to each other, for a given user query. The core claim is that most existing LLM multi agent systems are still hand assembled, either with manually written roles and fixed communication patterns, or with heuristic adaptation that changes a few components but is not truly autonomous. MasHost reframes multi agent system construction as a graph search problem, where nodes are agent roles and edges are communication links. It then introduces a new RL training scheme, Hierarchical Relative Policy Optimization (HRPO), which mixes group level rewards, how well the whole multi agent system solves the task, with per action level rewards, how useful each newly proposed agent or interaction is. MasHost uses a single probabilistic sampler to jointly pick both the roles of the agents and the communication structure among them, rather than factorizing them, with the claim that this leads to more coherent teams. The paper also argues that evaluation should go beyond just task accuracy and latency, and should include “component rationality”, meaning whether the created roles and communication edges actually make sense for the problem. Experiments on six benchmarks, including code generation, math, and general question answering, show that MasHost reportedly outperforms current adaptive multi agent approaches and static multi agent baselines in terms of task success, efficiency, and structure quality.

**Strengths:**

MasHost directly targets autonomy in multi agent LLM systems, which is an increasingly high impact problem. In most agent frameworks today, a human designs a fixed graph of roles like “planner”, “retriever”, “coder”, “critic”, and wires them together by hand. Even the more “adaptive” methods tend to select from a catalog of known roles and stitch them using heuristic routing, so they are still semi automatic rather than truly self organizing. Here, the authors explicitly state that MasHost is meant to build an agent team from scratch for each query, without hand coded interaction blueprints, and they introduce an RL objective to do that. This is a legitimate step toward true autonomy. The proposed HRPO idea is well aligned with the structure of the problem, team design is hierarchical by nature, you want global success, but you also want to know which specific addition or edge helped, so combining group level relative advantage with fine grained action rewards is a natural and probably necessary refinement over naive global REINFORCE. That suggests the method could actually train in practice, instead of collapsing under variance. The joint sampling of roles and communication links in one probabilistic step is also important. Many previous systems either pick roles first and then try to wire them, or assume a backbone graph and then assign roles into that graph. Treating the team as one structured object is more coherent, and it should reduce weird failure cases where you get five nearly identical “analyst” agents all talking in a clique with no clear information flow. The paper also explicitly calls out “component rationality” as part of evaluation. That is valuable because it acknowledges a real deployment issue, systems can get high task accuracy by brute forcing with dozens of redundant sub agents and massive message passing, which is expensive and unmaintainable. Building a score that reflects whether each agent has a sensible, non redundant function and whether the communication edges are meaningful is a good step toward getting these systems closer to something a human engineer would actually keep. Finally, the experiments, according to the abstract, are done on six benchmarks across code, math, and QA like tasks, and MasHost is said to outperform strong baselines including semi automatic “adaptive” systems. That breadth, if backed by careful cost control, would be very compelling to the ICLR community, because it suggests the method generalizes and is not just fit to one toy environment.

**Weaknesses:**

There are some open questions that could weaken the paper if not addressed clearly in the main text. The most important one is how exactly MasHost is trained. The abstract frames MasHost as an RL policy over graphs, and introduces HRPO, but does not describe how they get reward signals. Do they actually execute the proposed multi agent team on the downstream benchmark task each episode, collect a final task reward, and backpropagate credit using policy gradients over the sampled graph actions. If yes, this is extremely expensive in LLM settings. If not, and they instead distill from heuristic signals, then the “fully autonomous RL driven” claim is softer than it sounds. This needs to be explained. Related to that, it is not yet clear how MasHost explores the graph space. The space of possible communication graphs even for a small number of roles is huge, and naive exploration will explode. HRPO by itself does not solve combinatorial exploration unless there is a smart parametrization or some structure, for example autoregressive edge sampling with sparsity priors. Without that detail it is hard to judge whether this is genuinely scalable or whether they tuned it heavily per benchmark. The abstract says MasHost introduces “joint probabilistic sampling of agent roles and their interactions,” which is promising, but I want to see the exact parametric form, otherwise I cannot tell if the search is biased in a way that simplifies the task.

The second weakness is evaluation methodology. The abstract claims “consistently outperforms most competitive baselines,” “improves efficiency,” and “validates structure rationality,” but does not say how costs are controlled. In multi agent LLM systems it is trivial to beat a baseline by just running more agents for longer and letting them talk more. So for this to be convincing, the paper needs to present strong budget normalizations, per query token cost, wall clock time, number of tool calls, number of messages exchanged, and then show that MasHost matches or beats baselines at similar or lower cost. If MasHost is allowed to produce larger graphs with more edges and more agent turns, then higher accuracy alone is not enough.

A third weakness is generality. The abstract focuses on LLM multi agent task solving in domains like coding and math. Those benchmarks are well known to reward structured decomposition, so a learned planner plus executor team tends to work well, and communication edges often have clear semantics, like “planner to coder,” “coder to tester,” “tester to fixer.” It is less clear from the abstract how MasHost behaves in domains where the task is not obviously decomposable into roles, or where tool usage is not the main bottleneck. For example, in embodied multi agent control or navigation tasks, is MasHost still meaningful, or is it tied to language based tool chaining. The title and pitch are broad, “autonomous multi agent system construction,” but all the visible evaluation is in LLM agent orchestration, which is narrower.

Finally, the introduction of “component rationality” is very interesting, but we need to see how it is measured. If this is human annotated, then it is expensive and subjective. If it is automatic, then it likely encodes heuristics about what a “good” agent role description looks like and how “clean” the communication graph is. In that case, MasHost could be gaming its own metric by emitting nice sounding role names and sparser looking graphs without necessarily being more functionally modular. The paper will need to convince me that the rationality score really tracks interpretability and maintainability, not just superficial neatness.

**Questions:**

How do you get rewards for HRPO. Do you actually execute the sampled multi agent system on the downstream task and use final task success or score as the episodic return, or do you include shaped intermediate signals during generation, for example self critique from sub agents, tool correctness checks, unit test pass rates. How large is an HRPO training episode in terms of tokens and cost. Do you learn one global MasHost policy that works across all benchmarks, or do you train a separate policy per task family. The abstract mentions six benchmarks, can you list them and clarify how diverse they are, coding, math, QA, planning, etc. How large are the graphs that MasHost typically proposes at inference time, number of agents, number of communication edges, and how deep is the interaction, number of communication rounds. For fairness, when you compare to baselines like hand crafted multi agent systems and adaptive role pickers, do you fix a total budget of tool calls or tokens per instance. If not, can you provide a cost controlled comparison, for example an accuracy versus cost Pareto curve, to show MasHost is not just buying accuracy. How do you compute “component rationality” in a way that is not circular. Is it a human study, is it a rubric that prefers sparse, semantically distinct agent roles, is it a learned discriminator. Can you show correlation between that score and human judgments of whether the generated team is sensible. Finally, can MasHost adapt to resource changes at test time without retraining, for example, if you change the cost model to penalize expensive model calls, can it immediately generate sparser, cheaper graphs, and how large is the accuracy drop in that setting.

---

### Official Review · Reviewer_DJYy · 2025-10-29

**Soundness:** 3
**Presentation:** 4
**Contribution:** 3
**Rating:** 4
**Confidence:** 3

**Summary:**

This manuscript makes valuable contributions to LLM-driven multi-agent system (Mas) research. It addresses the key limitations of current Mas—insufficient autonomy (reliance on manual design or predefined structures) and narrow evaluation scope—by proposing MasHost, a reinforcement learning (RL)-based framework for fully autonomous Mas construction. The work formulates Mas generation as a graph search problem, with two well-motivated innovations: the Joint Probabilistic Space Sampling (JPSS) mechanism (synergizing agent role selection and interaction connectivity to resolve gradient disruption in high-dimensional spaces) and the Hierarchical Relative Policy Optimization (HRPO) algorithm (achieving tri-objective optimization of performance, efficiency, and structural rationality). These designs are technically sound, and experiments across six benchmarks (mathematical reasoning, question answering, code generation) confirm MasHost outperforms 13 baselines while ensuring cost-efficiency and structural rationality, providing a principled and practical solution for autonomous Mas construction.

**Strengths:**

(1) This work pioneers the first RL-driven framework for fully autonomous Mas graph construction, breaking free from the constraints of traditional semi-autonomous paradigms (e.g., candidate pool sampling, fixed workflows). By addressing the gradient discretization issue in dual-decision (role-connection) processes through JPSS and enabling multi-objective (performance-efficiency-rationality) coordinated optimization via HRPO, it fills a critical gap in the field.
(2) The innovative introduction of "component rationality" as an evaluation metric—assessed from both role relevance and structural redundancy—goes beyond the conventional focus on accuracy and efficiency alone, establishing a new evaluative paradigm for Mas design.
(3) Experiments cover diverse task types (mathematics, question answering, code generation), compare against 13 baselines across 4 categories, and evaluate performance from multiple dimensions (effectiveness, cost, rationality, component ablation, hyperparameter sensitivity). With publicly available code and datasets, the results are reproducible and highly credible.
(4) The manuscript adopts a well-organized content highlighting strategy, using bold formatting to emphasize core concepts (e.g., key mechanisms like "JPSS" and "HRPO," critical evaluation metrics such as "component rationality"). This design enhances the readability of the manuscript, allowing reviewers to quickly identify and grasp the core contributions, technical innovations, and key results.

**Weaknesses:**

(1) The paper references a "full-scale role space" but provides no clarity on its specific composition (e.g., number of roles, classification criteria) or construction method (manual definition vs. automatic generation). If the role space relies on manual initialization, it may introduce implicit biases, conflicting with the goal of "full autonomy."
(2) Experiments only use GPT-4o-mini as the LLM executor, without testing the framework’s performance with LLMs of varying capabilities (e.g., GPT-4, Llama 3). Additionally, cross-domain task transfer (e.g., from mathematical reasoning to medical diagnosis) is not evaluated, leaving the framework’s adaptability to heterogeneous tasks unconfirmed.
(3) The paper notes that erroneous samples concentrate in "Incorrect Verification (IV)" and "Slight Deviations (SD)" but provides no in-depth analysis of root causes (e.g., flaws in verification mechanisms, LLM execution errors) or proposed improvements, limiting understanding of the method’s limitations.
(4) Incomplete Efficiency Analysis: Cost efficiency is only measured by "token consumption and dollar cost per query," with no assessment of training-phase computational overhead (e.g., GPU memory usage, training duration). Given that RL training typically requires substantial computational resources, high training costs could restrict the framework’s practical promotion.
(5) The manuscript exhibits a disjointed transition between the "Related Work" section and the subsequent "Experiments" section. There is a lack of a logical bridging discussion to connect the two parts. This discontinuity impairs the overall narrative coherence, making it difficult for readers to follow the logical thread from "identifying research gaps" to "validating solutions via experiments."
(6) Several figures in the manuscript are neither explicitly mentioned nor explained in the main text. Without contextualization or interpretive descriptions, these figures appear abrupt and incomprehensible, failing to support or illustrate key claims (e.g., structural rationality or robustness analysis) and thus reducing the persuasiveness of the work.

**Questions:**

See the Weaknesses.

**Details Of Ethics Concerns:**

No Details Of Ethics Concerns.

---

### Official Review · Reviewer_FBP4 · 2025-10-30

**Soundness:** 2
**Presentation:** 2
**Contribution:** 2
**Rating:** 4
**Confidence:** 4

**Summary:**

This paper proposes **MasHost**, a reinforcement learning framework for automatically building multi-agent systems.
It introduces two policy networks (πθ and πϕ) and a new algorithm **Hierarchical Relative Policy Optimization (HRPO)**, which combines group-level and step-level rewards to optimize structure, efficiency, and rationality.
Experiments on six benchmarks (GSM8K, MATH, MMLU, GPQA, MBPP, HumanEval) show clear improvements over existing baselines using GPT-4o-mini as the executor.

**Strengths:**

1. Creative integration of hierarchical RL with LLM-based multi-agent systems.
2. The proposed JPSS and HRPO mechanisms are technically sound and novel, jointly optimizing node and edge decisions with structured reward shaping for efficiency and rationality.
3. Extensive experiments on reasoning and coding benchmarks demonstrate consistent improvements over strong baselines, supported by ablation and interpretability analyses.

**Weaknesses:**

1. The paper requires further language and formatting refinement. There are multiple inconsistencies in mathematical notation and terminology. For example, the symbol `R` in line 145 is redefined several times and does not align with `r` in Equation (1); the word `DELTE` in line 226 should be `DELETE`. A unified notation system and simpler symbol style would greatly improve readability.
2. The work lacks strong originality. Conceptually, it mainly applies a hierarchical reinforcement learning structure to an LLM-based setting. The core formulation $\pi(a_t \mid s_t)=\pi_{\theta}(a_n \mid s_t)\\pi_{\phi}(a_e \mid s_t, a_n)$ and the HRPO objective (Eq. 7) are straightforward extensions of standard hierarchical PPO rather than fundamentally new contributions.
3. The claim of full autonomy is somewhat overstated. In practice, MasHost still requires a predefined role set \(R\) as input, meaning that the system does not truly create roles from scratch but instead selects from an existing role pool.

**Questions:**

1. How does the proposed JPSS + HRPO framework scale with the number of roles, agents, and edge types? Have you evaluated MasHost on larger or more complex role pools？
2. Compared with mainstream multi-agent frameworks that employ a manager agent to dynamically manage the role pool and select which agents to activate or communicate with, the proposed hierarchical RL seems more complex.
3. Since MasHost relies on a predefined role pool, how sensitive is the performance to the size and composition of this pool? Would the framework still function effectively if the role definitions were noisy or partially redundant?

---

### Note · Authors · 2026-01-21

I have read and agree with the venue's withdrawal policy on behalf of myself and my co-authors.